# First Report of Porcine Parvovirus 8 in Europe: Widespread Detection and Genetic Characterization on Commercial Pig Farms in Hungary and Slovakia

**DOI:** 10.3390/ani14131974

**Published:** 2024-07-03

**Authors:** Barbara Igriczi, Lilla Dénes, Kitti Schönhardt, Gyula Balka

**Affiliations:** 1Department of Pathology, University of Veterinary Medicine, István Str. 2., 1078 Budapest, Hungary; igriczi.barbara@univet.hu (B.I.); denes.lilla@univet.hu (L.D.); schonhardt.kitti@univet.hu (K.S.); 2National Laboratory of Infectious Animal Diseases, Antimicrobial Resistance, Veterinary Public Health and Food Chain Safety, University of Veterinary Medicine, István Str. 2., 1078 Budapest, Hungary

**Keywords:** porcine parvovirus 8, novel porcine parvoviruses, first detection, viral prevalence, oral fluid

## Abstract

**Simple Summary:**

Porcine parvoviruses (PPVs) are small DNA viruses that are widespread in swine populations. Recently, a novel parvovirus, PPV8, was identified in China. This study aimed to detect the presence of PPV8 in Europe, specifically in Hungarian and Slovakian pig farms. A large number of samples, including blood serum, oral fluids, and processing fluids were analyzed, revealing the presence of the virus in 68% of the farms included in our study. The newly identified PPV8 strains were highly similar to the Chinese strain, but the amino acid differences might indicate local evolution of the virus.

**Abstract:**

Porcine parvovirus 8 (PPV8), a novel virus in the *Parvoviridae* family, was first identified in 2022 in lung samples of domestic pigs from China. Retrospective analyses showed that it had been circulating in China since 1998, but no other countries had reported its presence so far. A recent study conducted in South America did not detect any PPV8-positive samples in that region. Here, we report the detection of PPV8 in Hungarian and Slovakian pig farms and the estimated prevalence of the virus in Hungary. Altogether, 2230 serum, 233 oral fluid, and 115 processing fluid samples were systematically collected from 23 Hungarian and 2 Slovakian pig farms between 2020 and 2023. A real-time quantitative PCR method was developed to detect the viral genome. Our results revealed the presence of PPV8 on 65% of the Hungarian farms and both Slovakian farms included in our study, marking its first detection in Europe. Oral fluid samples showed the highest positivity rates, reaching up to 100% in some herds. The viral genome was successfully detected in serum and processing fluid samples too, but with significantly lower prevalence rates of 4% and 5%, respectively. Genetic analysis of 11 partial VP2 sequences demonstrated high similarity to the original Chinese strain but with unique amino acid mutations, suggesting possible local evolution of the virus. Our study presents the first scientific evidence of PPV8 infection outside of China and offers a comprehensive assessment of its prevalence in the Hungarian pig population. Further research is required to understand its potential impact on swine health.

## 1. Introduction

Porcine parvovirus 8 (PPV8, species *Protoparvovirus ungulate 4*), a novel member of the *Parvoviridae* family, was first detected in 2022 by high-throughput sequencing (HTS) of lung tissue samples originating from PRRSV-positive pigs in China [1]. Parvoviruses are small, non-enveloped viruses with linear, single-stranded DNA genomes, typically ranging from 4 to 6.3 kilobases in length. Their genome contains two major open reading frames (ORFs) that encode for the non-structural proteins (NS) and capsid proteins (VP1/VP2), crucial for viral replication and structure, respectively [2].

The *Parvoviridae* family is divided into three subfamilies: *Parvovirinae* and *Hamaparvovirinae*, which infect vertebrates, and *Densovirinae*, which infect arthropods. To date, eight different parvoviruses have been identified in pigs (PPV1–8), which are classified into the *Protoparvovirus*, *Tetraparvovirus*, *Copiparvovirus*, and *Chaphamaparvovirus* genera [3].

The first known PPV (PPV1) was identified in the 1960s and remains a major cause of reproductive losses in pigs [4]. Recent advances in metagenomic technology have led to the identification of novel PPVs worldwide [1,5,6,7,8,9,10]. Recent studies on PPV2–PPV7 indicate that these viruses are highly prevalent in the pig populations worldwide [11,12,13,14,15,16,17,18,19]. However, unlike PPV1, their pathogenic roles and clinical relevance have not been defined. These novel PPVs are frequently detected in coinfections with various pathogens, such as porcine circovirus type 2 (PCV2) or porcine reproductive and respiratory syndrome virus (PRRSv) [13,14,18,20,21], potentially making the clinical conditions more severe.

Genomic comparisons show that PPV8 shares 44.18% nucleotide identity with PPV1, but only 16.23–24.17% similarity with PPV2–7, indicating significant genetic diversity among novel PPVs. Despite its relatively low sequence similarity to other PPVs, PPV8 retains conserved amino acid sites that are characteristic of parvoviruses [1]. Phylogenetic analysis based on the NS1 gene further supports that PPV8 is significantly distinct from other known novel parvoviruses, as it shares the highest sequence similarity with PPV1 and clusters with Protoparvoviruses [1]. According to the latest ICTV classification criteria for parvoviruses, it qualifies as a new species within the *Protoparvovirus* genus [3].

The first retrospective study of PPV8 suggests that the virus has a wide geographical distribution and long-term existence in China [1], but it has not been reported outside of this country. A recent study in Colombia found no PPV8-positive cases in 234 serum samples of gilts collected from 40 different, clinically healthy herds [19]. In the present study, our aim was to screen large-scale pig herds in Hungary and Slovakia to detect the presence of PPV8.

## 2. Materials and Methods

### 2.1. Sample Collection

Serum, oral fluid, and processing fluid samples were gathered between 2020 and 2023, from 23 Hungarian and 2 Slovakian pig farms to screen for various pathogens (Table 1). Samples were systematically collected on farrow-to-finish farms using a comprehensive cross-sectional sampling protocol. The farms varied in basic production parameters, genetics, and herd size but vaccinations against PPV1 were performed on all of them. For sows, the population ranged from 520 and 2200, while for gilts, the numbers were between 72 and 1300, for weaned pigs, between 1270 and 9000, and for fatteners, they varied from 2164 and 17,500. No outbreaks or overt clinical symptoms were reported in the herds during the sampling period. From each herd, 100 serum samples were collected from pigs of different age groups (2-, 4-, 6-, 8-, 10-, 14- and 18-week-old pigs, gilts, and sows of 2 and 4 parities). Additionally, 10 pen-representative oral fluid samples were collected by sampling ropes: 5 samples were obtained from 8–12-week-old, weaned pigs and 5 samples from 18–20-week-old fatteners. Finally, 5 processing fluid samples were collected during piglet castration, which includes the testicles of approximately 10 litters per sample. Table 1 summarizes the number of collected samples, which varied in some cases.

In total, 2230 serum samples, 233 oral fluid, and 115 processing fluid samples were obtained, and all were stored at −80 °C until processing and further analysis. The farms’ participation in this sampling campaign was voluntary, with no significant clinical diseases reported during the period of samplings. More details of the sampling strategy can be found in our previous studies [22,23].

### 2.2. Sample Processing and DNA Extraction

Prior to DNA extraction, the processing fluid and oral fluid samples were centrifuged (300× *g* for 5 min at room temperature) and the serum samples belonging to the same age group were pooled in groups of 5. The viral DNA was extracted from oral fluid, processing fluid, and serum samples with the QIAmp cador Pathogen Mini Kit (Qiagen, Hilden, Germany) using the QIAcube automatic nucleic acid extractor according to the manufacturer’s protocol. Nucleic acids were stored at −80 °C until further analysis.

### 2.3. qPCR Detection of PPV8

Real-time quantitative PCR (qPCR) was used to detect the presence of viral DNA in the examined samples. The PCR assays were run on a Q qPCR Machine (Quantabio, Beverly, MA, USA), and each reaction contained 10 µL PerfeCTa qPCR ToughMix (Quantabio, Beverly, MA, USA), 2 µL extracted DNA, 900 nM specific primers (forward: 5′-GCATGATGCCATACACACC-3′ and reverse: 5′-TGTCTTGTTGCTTGTCCTTG-3′), and 250 nM probe 5′-HEX-TGGAACCCTTTCGTTCCTCCAATCTACAA-BHQ-3′) in a 20 µL final volume. The primers and probe were designed to detect the VP2 capsid gene using the Eurofins PCR Primer Design Tool (https://eurofinsgenomics.eu/en/ecom/tools/pcr-primer-design/, accessed on 5 February 2024). The reference sequence used for primer design was the first available PPV8 sequence in GenBank, with accession number: OP021638. For the PCR reactions, we used the following temperature profile: 95 °C for 3 min followed by 40 cycles of 95 °C for 10 s and 60 °C for 30 s.

Statistical analysis of the qPCR results was conducted with GraphPad Prism 8. Fisher’s exact test was used to compare the prevalence of PPV8 across different diagnostic matrices. The Ct values detected in different sample types were analyzed with the Mann–Whitney test through pairwise comparison. Statistical significance was set at *p* < 0.05.

### 2.4. Genetic Analysis

Sequencing PCR reactions were carried out according to the PCR method described by Guo et al. 2022 [1]. In each reaction, 5 µL extracted DNA was added to a mix of 5× Phusion^TM^ HF Buffer (Thermo Scientific^TM^), 200 µM dNTPs, 1 µM of each primer (PPV8-inF: 5′-TCCAAGTTGCCCTAGACAGC-3′ and PPV8-inR: 5′-GCCTCGTACATGTGGACCTC-3′), and 0.5 units of Phusion^TM^ High-Fidelity DNA Polymerase. The reactions were run in a Genesy 96T gradient PCR machine (Tianlong, Xi’an, China) with the following temperature profile: 98 °C for 30 s followed by 45 cycles of 98 °C for 10 s, 65 °C for 20 s, and 72 °C for 45 s followed by a final elongation step at 72 °C for 5 min. After agarose gel electrophoresis, the 554 bp long amplicons were manually cut and then purified using Qiagen Gel Extraction Kit (Qiagen, Hilden, Germany) according to the manufacturer’s instructions. Capillary electrophoresis was conducted by a commercial provider, Eurofins BIOMI Kft. (Gödöllő, Hungary). All chromatograms were visualized and trimmed using Chromas 2.6.6 software (Technelysium Pty Ltd., South Brisbane, Australia). The forward and reverse sequences were aligned and assembled using the E-INS-I method of the online software MAFFT version 7 [24]. The obtained sequences were compared with each other and a reference strain downloaded from GenBank (OP021638).

A comprehensive alignment was conducted against different parvoviruses (mostly PPV1–PPV8). The reference VP2 sequences used for the phylogenetic analysis and tree reconstruction were obtained from GenBank. The phylogenetic tree was constructed by the neighbor-joining method with 1000 replicates of bootstrap analysis, using MEGAX software [25]. The Interactive Tree Of Life (iTOL) version 6 online tool [26] was used for displaying and editing the phylogenetic tree. All PPV8 sequences detected in this study have been deposited in the NCBI GenBank under the accession numbers PP781989–PP781999.

## 3. Results

A total of 15 out of 23 Hungarian farms (65%) and both of the Slovakian farms, which were located close to the Hungarian border, were positive for PPV8 (Figure 1). We considered any farm positive where the viral genome was detected in at least one sample of a minimum of one diagnostic matrix. The prevalences within the herd varied significantly among the PPV8-positive farms, as summarized in Table 1.

For the following prevalence estimation, only the Hungarian samples were included. Comparing the different diagnostic matrices, oral fluid samples exhibited notably high detection rates (Figure 2a), ranging from 33.3% to 100% in positive herds. Overall, 46% (101/218) of the oral fluid samples were PPV8-positive. The mean Ct values for weaned pigs (8–12 weeks old) were 31.61 ± 3.22, while those for fatteners (18–20 weeks old) were 32.56 ± 3.47. No significant differences were observed between the viral prevalence, among weaned pigs and fatteners, as 45% (51/114) of weaned pigs and 48% (50/104) of fatteners were PPV8-positive. We also detected the virus in 4% (15/412) of serum pools. The prevalence within the herd ranged between 5% and 30%, with mean Ct values of 31.79 ± 2.56. The most affected age groups were 10-week-old pigs with an 8% PPV8-positivity rate, and 8- and 14-week-old pigs, each with a 7% positivity rate. PPV8-positive processing fluids were only found in one herd, where all five samples tested positive, resulting in an overall prevalence of 5% (5/106). The Ct values of this sample type were significantly higher (35.87 ± 1.24) than those detected in serum samples or oral fluids (Figure 2b).

In Slovakia, we also detected PPV8 viral DNA in three oral fluid samples (one from Farm 24 and two from Farm 25) in total. The other diagnostic matrices were all PPV8-negative.

We have successfully obtained 554-bp-long partial VP2 sequences from four serum and seven oral fluid (six Hungarian and one Slovakian) samples. The initial identification of the obtained sequences using the NCBI BLAST system showed one match, which was the only existing PPV8 sequence to date. The partial VP2 sequences obtained in this study were compared with each other and the corresponding section of the Chinese PPV8 strain GDJM2021, as summarized in the pairwise nucleotide identity matrix in Figure 3a. All Hungarian sequences displayed 98.02–99.10% nucleotide identity compared to the strain GDJM2021. Similarly, the Slovakian strain showed high similarity to the Chinese (99.1%) and the Hungarian sequences also (97.83–99.64%). Two amino acid changes were identified in the examined section of the capsid gene. Starting from the VP2 gene’s start codon, the K256Q mutation was found in three strains and the V274I mutation was present in two strains (Figure 3b). The origins of these sequences are indicated on the map with blue and red circles, respectively (Figure 1).

A phylogenetic tree was constructed using a set of reference VP2 sequences from all PPVs and various parvovirues from different organisms (Figure 3c). This evolutionary tree shows eight clusters, each representing one of the eight different PPVs and related parvoviruses within the same genus. All 11 of our PPV8 sequences grouped with the Chinese strain GDJM2021, branching off but remaining within the same clade as PPV1 representing the Protoparvovirus genus. Our PPV8 sequences exhibit 28.16–32.94% nucleotide identity with other members of the Protoparvovirus genus and only 22.09–31.76% intercluster homology with the novel PPVs (PPV2–PPV7).

## 4. Discussion

In recent years, the incidence of emerging infectious diseases in pigs has been increasing, spreading rapidly due to global trade and the movement of people and animals. Although the impact of these newly identified viruses on swine health is still largely unknown, detecting and assessing their prevalence in pig populations can help us understand the significance and epidemiology of emerging viruses. The aim of this study was to investigate the presence of PPV8 infection in our sample collection. For rapid and specific detection of the PPV8 viral DNA, we developed a probe-based qPCR method and examined serum, oral fluid, and processing fluid samples collected from 23 Hungarian and 2 Slovakian pig farms.

The viral genome was successfully detected in 65% of the Hungarian farms, and both Slovakian herds. Our results indicate that PPV8 is widespread in the Hungarian pig population and the virus is also present in Slovakia. According to our previous results, it can be stated that among the diagnostic matrices used in our study, all novel PPVs, including PPV8 are most commonly detected in oral fluid samples and have the lowest prevalence in processing fluids. These results align with a similar study conducted in Poland, where high detection rates of novel PPVs were observed in oral fluid samples during the examination of large numbers of serum, oral fluid, and fecal samples [17]. The high positivity rate in oral fluids can be explained by their pen-representative nature, collected from weaned pigs and fatteners across different production units. Although this sampling method increases the likelihood of contamination, the ease of collection and ability to represent a collective status make oral fluids suitable for herd- or group-level surveillance. Despite representing multiple animals, some PPV8-positive oral fluid samples were found with notably lower Ct values, compared to other samples, indicating higher viral concentrations in these cases (Figure 2b).

The detection of the virus in 4% of serum pools indicates active viremia in a small percentage of the animals during the sampling. The majority of the PPV8-positive serum samples were obtained from weaned pigs and fatteners, which aligns with the ages of the animals from which the oral fluid samples were collected. Viral DNA was successfully detected in processing fluid samples too, representing the youngest age group in our study. The percentage of the PPV8-positive samples was low (5%), but it might suggest congenital infections in the herd with the positive cases.

Although the classification of the parvoviruses is based on the NS1 sequence of the viral genome, our evolutionary tree based on the VP2 sequence successfully represents the genetic relationships between the PPVs. All PPVs were distinct from each other and clustered together with other parvoviruses of the same genus. The phylogenetic analysis of the Hungarian and Slovakian PPV8 strains showed that our sequences cluster with the only Chinese PPV8 strain and are distinct from other PPVs. They belong to the Protoparvovirus genus, showing a close genetic relation to PPV1 and other parvoviruses belonging to this viral genus.

Genetic analysis of the partial VP2 sequences revealed high nucleotide- and amino acid identity compared with the original Chinese strain, with only two specific amino acid mutations in some Hungarian strains. The geographically clustered farms where these mutations were found are located in the southwestern part of Hungary. Since these amino acid changes occur in the VP2 capsid gene, they might affect the receptor binding and antigenic properties of the virus. Further research is required to determine the virus’ exact structure, virulence, and impact on the host organism.

## 5. Conclusions

Our study presents the first scientific evidence of PPV8 infection outside of China and confirms the presence of the virus in Europe, specifically within Hungarian and Slovakian swine herds. The samples were collected between 2020 and 2023, which means that PPV8 has been circulating in this region for at least four years. Since the examined herds reported no clinical symptoms or major disease outbreaks at the time of the sampling, conclusions regarding the pathogenicity of PPV8 cannot be drawn. Further research is needed to assess the virus’s prevalence in Europe and to understand its impact on swine health.

## Figures and Tables

**Figure 1 animals-14-01974-f001:**
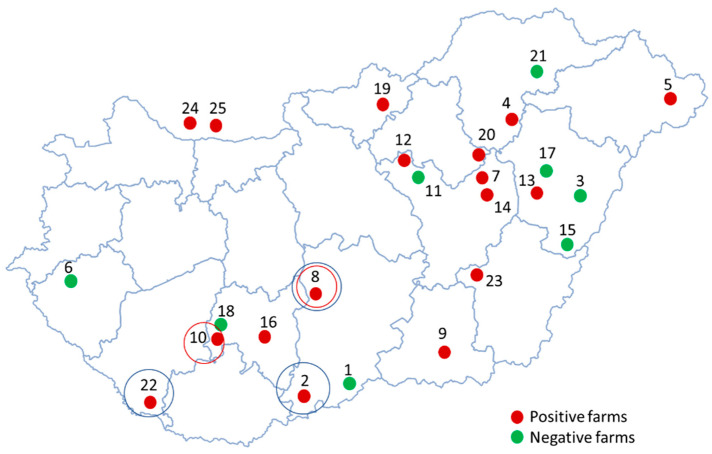
Map of Hungary showing the geographical location of the 23 examined Hungarian and 2 examined Slovakian farms located close to the northern border. The green dots represent the PPV8-negative farms and the red dots represent the PPV8-positive farms where at least one positive sample was found. The blue and red circles around the farms indicate the origin of strains with K256Q and V274I amino acid mutations, respectively.

**Figure 2 animals-14-01974-f002:**
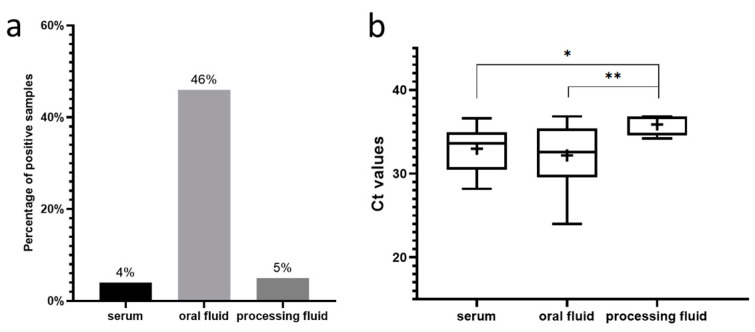
(**a**) Percentages of PPV8-positive serum pools, oral fluid, and processing fluid samples collected in Hungary (**b**) Boxplots representing the Ct values of the PPV8-positive serum pools, oral fluid, and processing fluid samples collected in Hungary. The whiskers show the minimum and the maximum, and the average values are indicated with “+” signs. The horizontal lines of the box show the first quartile, the median, and the third quartile. The asterisks above the columns represent the statistically significant differences (*: *p* < 0.05; **: *p* < 0.01).

**Figure 3 animals-14-01974-f003:**
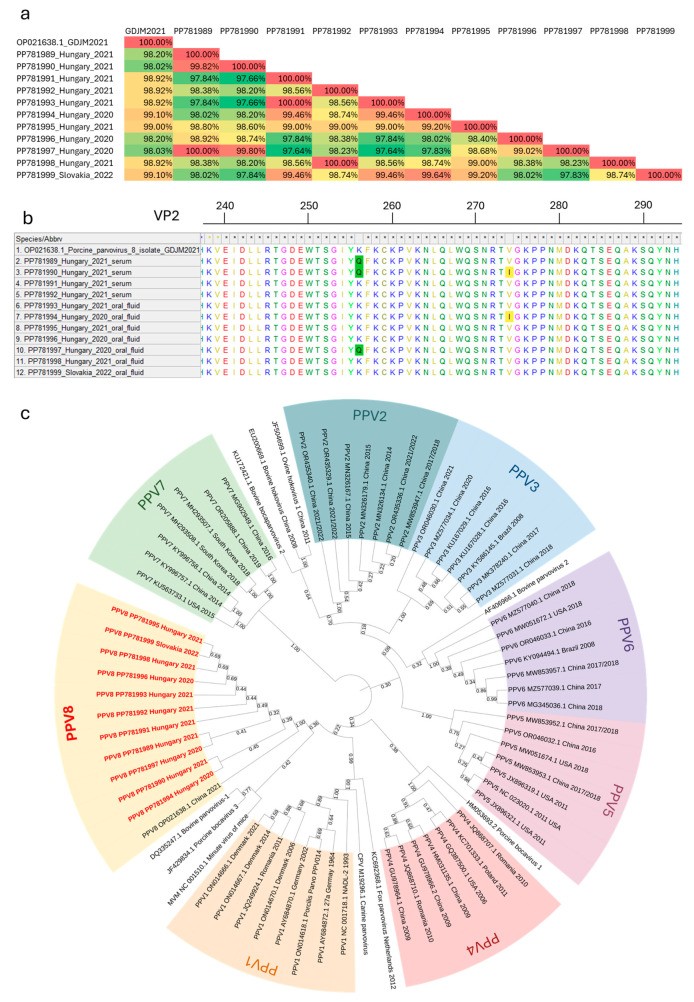
(**a**) Matrix showing the pairwise nucleotide identity measures of all PPV8 sequences. Each cell shows the percentage of nucleotide identity between two sequences, with higher percentages indicating greater similarity. The diagonal values represent the self-comparisons, which are 100%. (**b**) Amino acid sequence alignment of the PPV8 strains detected in this study, highlighting the two amino acid mutations, compared to the PPV8-GDJM2021 strain. (**c**) Phylogenetic tree of PPVs. This evolutionary tree was constructed in MEGAX, using 61 reference VP2 sequences of different parvovirus strains. Sequences of all eight PPVs are color-coded and the PPV8 sequences identified in this study are highlighted in red.

**Table 1 animals-14-01974-t001:** Summary of the examined Hungarian and Slovakian farms, sample types and sample sizes, and the number and percentage of PPV8-positive samples.

Hungarian Farms	Serum Samples	Serum Pools	PPV8- Positive	Oral Fluids	PPV8-Positive	Processing Fluids	PPV8- Positive
Farm 1	100	20	0	10	0	5	0
Farm 2	100	20	3 (15%)	10	9 (90%)	4	0
Farm 3	100	20	0	12	0	10	0
Farm 4	0	0	0	10	9 (90%)	3	0
Farm 5	100	20	0	10	10 (100%)	5	0
Farm 6	100	20	0	10	0	8	0
Farm 7	90	18	0	10	9 (90%)	1	0
Farm 8	130	26	2 (7.7%)	10	5 (50%)	2	0
Farm 9	100	20	1 (5%)	10	5 (50%)	5	0
Farm 10	100	20	1 (5%)	10	8 (80%)	5	0
Farm 11	80	16	0	8	0	5	0
Farm 12	100	20	0	9	3 (33.3%)	5	0
Farm 13	100	20	2 (10%)	10	6 (60%)	5	5 (100%)
Farm 14	60	12	0	10	10 (100%)	5	0
Farm 15	100	20	0	10	0	5	0
Farm 16	70	14	0	5	2 (40%)	0	0
Farm 17	100	20	0	10	0	5	0
Farm 18	60	12	0	4	0	4	0
Farm 19	100	20	0	10	5 (50%)	5	0
Farm 20	100	20	0	10	10 (100%)	3	0
Farm 21	70	14	0	10	0	2	0
Farm 22	100	20	0	10	5 (50%)	5	0
Farm 23	100	20	6 (30%)	10	5 (50%)	9	0
Total	2060	412	15	218	101	106	5
Slovakian farms							
Farm 24	100	20	0	10	1 (10%)	5	0
Farm 25	70	14	0	5	2 (40%)	4	0
Total	170	34	0	15	3	9	0

## Data Availability

Sequence data gathered during the study can be accessed under GenBank accession numbers: PP781989–PP781999. Data regarding the name and exact location of the farms involved in the study are confidential due to the commercial rights of the owners.

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
