# Peer review of "First Report of Porcine Parvovirus 8 in Europe: Widespread Detection and Genetic Characterization on Commercial Pig Farms in Hungary and Slovakia"

_animals, 2024, doi:10.3390/ani14131974_

Round 1
Reviewer 1 Report
Comments and Suggestions for Authors
This study reports the prevalence of PPV8 in pig farms in Hungary and Slovakia, which is important for the prevention and control of PPV8. However, this paper has a rough data analysis and overall logic, while the discussion part is scarce, and it is recommended to revise the whole paper. The following are some of the contents of this manuscript detailed questions.
1、Oral fluid samples showed the highest positivity rates, reaching up to 100% in some herds. it should be stated in detail how much its positive rate is.
2. Some writing problems, please correct. For example, 1. Figure should be Figure 1 and so on.
3、Figure 1,the position of the red and green dots is based on what calibration, found that the red and green dots are marked outside the map, at the same time, two dots of the same colour appear in one place. Please explain in detail or describe in detail with a chart in the attachment.
4, Add a result about VP2 evolutionary tree analysis as well as homology analysis.
5. In the article, only part of the VP2 gene was sequenced, how come the result about the mutation of VP1 gene came out.
This study reports the prevalence of PPV8 in pig farms in Hungary and Slovakia, which is important for the prevention and control of PPV8. However, this paper has a rough data analysis and overall logic, while the discussion part is scarce, and it is recommended to revise the whole paper. The following are some of the contents of this manuscript detailed questions.
1、Oral fluid samples showed the highest positivity rates, reaching up to 100% in some herds. it should be stated in detail how much its positive rate is.
2. Some writing problems, please correct. For example, 1. Figure should be Figure 1 and so on.
3、Figure 1,the position of the red and green dots is based on what calibration, found that the red and green dots are marked outside the map, at the same time, two dots of the same colour appear in one place. Please explain in detail or describe in detail with a chart in the attachment.
4, Add a result about VP2 evolutionary tree analysis as well as homology analysis.
5. In the article, only part of the VP2 gene was sequenced, how come the result about the mutation of VP1 gene came out.
Reviewer 2 Report
Comments and Suggestions for Authors
The objective of this paper was to detection of PPV8, previously detected in China, in Europe, on Hungarian and Slovakian farms. Currently, well known PPV virus is still dangerous pathogen, but we have vaccine to protect pigs. Porcine parvovirus type 8 is a newly, probably emerging pathogen, but lack of commercially available vaccine.
Major comments:
Linje 80 - More details of the sampling strategy can be found in our previous studies [12], [13]. Please add more details, thus both references are cited as self-citation
2.1 Sample collection – please add information about number of animals on farms. Only information is about sow herd size. Moreover add more details about sampling – from 1 animal was taken only serum, oral fluid or both?
2.2 Sample processing and DNA extraction – please add centrifuge condition and methods used to estimate DNA concentration and purity
2.3 qPCR detection and genetic analysis of PPV8 – please add reference sequence using to primer design and positive and negative control. Therefore, you cannot tell that the observed curves are PPV8.
Which primers were used perform PCR to sequencing? Moreover, how many samples were sequenced and from which type of samples? . If Authors successfully obtained 554 bp long partial VP2 sequences, both positive and negative controls should be added and primers sequences.
Line 118 - the alignment against the reference PPV8 sequence… - please add number of this sequence
3. Results and Discussion must be separated.
Line 163 - According to our previous results, it can be stated that all novel PPVs, including 1 PPV8 are most commonly detected in oral fluid samples and have the lowest prevalence 1 in processing fluids (manuscript under review). Authors should be added reference or add comments unpublished data
Most of the cited articles are outdated. 9 from 15 are older than 5 years.
References should be described as follows, depending on the type of work:
Journal Articles:
- Author 1, A.B.; Author 2, C.D. Title of the article. Abbreviated Journal Name Year, Volume, page range.
Round 2
Reviewer 2 Report
Comments and Suggestions for Authors
The authors addressed the major comment; however, minor corrections should be considered as follows:
The reference sequence used for primer design was the only available PPV8 sequence in GenBank, with accession number: OP021638. Please revised sentence, now 5 PPV8 complete genome sequence are available (GenBank: PP842648.1, PP842647.1, PP842646.1, PP842645.1, PP842644.1)